# ER-LAC: Span-Based Joint Entity and Relation Extraction Model with Multi-Level Lexical and Attention on Context Features

Yaqin Zhu, Xuhang Li, Zijian Wang *, Jiayong Li, Cairong Yan and Yanting Zhang

School of Computer Science and Techonology, Donghua University, Shanghai 201620, China; zhuyaqin@mail.dhu.edu.cn (Y.Z.); lixuhang@mail.dhu.edu.cn (X.L.); lijiayong@mail.dhu.edu.cn (J.L.); cryan@dhu.edu.cn (C.Y.); ytzhang@dhu.edu.cn (Y.Z.)
* Correspondence: wang.zijian@dhu.edu.cn

**Abstract:** In recent years, joint entity–relation extraction (ERE) models have become a hot research topic in natural language processing (NLP). Several studies have proposed a span-based ERE framework, which utilizes simple span embeddings for entity and relation classification. This framework addresses the issues of overlap and error propagation that were present in previous entity–relation extraction models. However, span-based models overlook the influence of lexical information on the semantic representation of the span and fail to consider relations with a strong intrinsic connection between span pairs. To tackle these aforementioned issues, we present a new ERE model called ER-LAC (Span-based Joint Entity and Relation Extraction Model with Multi-level Lexical and Attention on Context Features). This model is designed with multi-granularity lexical features to enhance the semantic representation of spans, and a transformer classifier is employed to capture the internal connections between span pairs, thereby improving the performance of relational classification. To demonstrate the effectiveness of the proposed model, ablation experiments were conducted on the CoNLL04 dataset. The proposed model was also compared with other models on three datasets, showcasing its computational efficiency. The results indicate that the introduced lexical features and classifier enhance the F1 score for entity extraction by 0.84% to 2.04% and improve the F1 score for relationship classification by 0.96% to 2.26% when compared to the previous state-of-the-art (SOTA) model and the baseline SpERT model, respectively.

**Keywords:** named entity recognition; relation extraction; lexical features; internal connections; natural language processing



## 1. Introduction

Entity and relation extraction (ERE) is the process of automatically identifying entities and the relations between them from natural language text. This task includes two subtasks: named entity recognition (NER) and relation extraction (RE). ERE plays a crucial role in various applications [1]. In practical applications, entity–relation extraction techniques have been widely used in knowledge graphs, question-and-answer systems, information retrieval, intelligent customer service, and other fields. Traditional supervised entity–relation extraction methods are mainly divided into feature engineering-based methods and kernel function based methods [2,3]. However, these methods rely on manual feature extraction, which can lead to issues such as error propagation and limited effectiveness in capturing long-tailed entities and complex relations. In recent years, there has been a surge of research on entity–relation extraction methods based on deep learning. These methods have demonstrated consistent improvements in performance across multiple datasets [4–7].

Two commonly used categories of deep learning-based supervised methods for entity–relation extraction are pipelined learning and joint learning. In pipelined learning, the entity extraction task is performed first using approaches based on recurrent neural networks

(RNNs), convolutional neural networks (CNNs), or long short-term memory (LSTM). Subsequently, the relationship extraction task is carried out. For instance, MV-RNN [8] was the first to propose using RNN for entity–relation extraction, building semantic vector representations of words and phrases based on parse trees. However, this approach relies on the syntax tree used in the process, and errors in syntactic analysis can affect the final results of the model. Yan et al. [9] proposed a method for entity–relation extraction based on LSTM by combining the shortest path of dependency analysis with word vector features and lexical features. However, all of these methods encounter the issue of error propagation, wherein an error in entity recognition can significantly impact the subsequent relation extraction. Furthermore, the interdependence between the two subtasks of entity extraction and relation extraction is frequently overlooked.

In contrast, the joint ERE model offers a specific methodological framework for addressing ERE tasks. This model leverages shared features and inherent connections between the subtasks of entity extraction and relation extraction. By doing so, it efficiently and effectively extracts entities and relations from textual data, resulting in improved performance [4]. This approach can also avoid problems with error propagation. Deep learning models based on sequence labeling frameworks are a common method for joint extraction of entity and relations. However, they are unable to identify overlapping entities, which are frequently found in natural language [5,10]. To overcome this limitation, pointer network-based approaches have been proposed, which use multiple pointer networks for multi-sentence annotation and multiple tags to represent a sentence [11,12]. However, such an approach may suffer from label imbalance. Therefore, more recent methods are attempting to use novel span-based entity–relation joint extraction models to handle ERE tasks [13–15].

In the span-based joint entity–relation extraction method, word spans in a sentence are enumerated, and the representation embedding of spans is shared in one model to accomplish the tasks of NER and RE. This method overcomes issues such as overlapping entity recognition and repeated encoding in entity–relation joint extraction models [13–15]. For this method, named entity recognition is regarded as a classification task for each word span sample, and relation extraction is treated as a relation classification task for the combinations of spans and context embeddings. Fundamentally, the entity–relation joint extraction task is transformed into an entity and relation classification task based on span representation embedding in the method. Therefore, the representation embedding of spans has a significant impact on the model performance. Previous studies have optimized the span embedding representation to improve the performance of the model on entity and relation extraction [1,6,7,13,14,16–18]. For instance, Dixit et al. [13] and Luan et al. [1,6] used Bi-LSTM to obtain global span representation, which enhanced span representation richness by encoding information of bidirectional sequential context in a span, and improved model effectiveness. DyGIE++ [7] and SpERT [14] used pre-training models to enhance span representation, fused span representation based on pre-training models trained on large-scale corpus, and made the span representation have more extensive and accurate semantic expression. However, most studies on span representation embedding currently ignore the multi-level grammatical structure information in text. Most multi-span representations used for relationship classification also adopted simple concatenation and lack detailed semantic expression of contextual words in spans. At the same time, although the pre-training models with an attention mechanism have been used for span-based joint entity–relation extraction methods to obtain initial word embeddings, existing models still lack dynamic attention weighting for complex span or span pair embeddings in entity recognition and relation extraction tasks. These issues significantly limit the performance of the span-based joint entity–relation extraction models.

To address the aforementioned issues, the span-based joint entity and relation extraction model with multi-level lexical and attention on context features (ER-LAC) is proposed in this paper. In this model, multi-level lexical information, including coarse-grained and fine-grained lexicon features, is integrated into the span embedding representation for

both NER and ER tasks. Coarse-grained and fine-grained lexical features are encoded at the span and token levels, respectively, and combined with the original span embeddings to form enriched span embeddings with lexical information. Furthermore, multi-head self-attention modules [19] are employed in both sub-tasks, dynamically weighting span representations based on the intrinsic correlations among different feature dimensions of the spans, to improve the performance of the model on named entity recognition and relation extraction. The proposed model has been comprehensively evaluated on three commonly used entity–relation extraction datasets, and it is observed that the proposed model achieves 0.84% and 0.96% F1 score improvements over the SOTA models in NER and RE sub-tasks, respectively. The main contributions of this work are listed as follows:

1. A multi-level lexical feature embedding method enhanced for span embedding representation was proposed, which enhances the semantic expression ability of span embedding representation in entity recognition and relation extraction through coarse-grained and fine-grained lexical feature embedding encoding at the span level and token level.

2. A relation extraction submodule based on a transformer encoder structure with multi-head self-attention was proposed, which can enhance the model's relation extraction performance by extracting temporal information of words in span and allocating self-attention weights accordingly.

3. The proposed model was found to outperform other SOTA models in three commonly used entity–relation extraction datasets.

In the remaining part of the paper, the related works that are relevant to the proposed method in this paper are introduced in Section 2. Then, in Section 3, a detailed description of the proposed method is provided. The experimental setup and results are presented and then a discussion of the method and experimental results is given in Section 4. Finally, in Section 5, the conclusion of this work is given.

## 2. Related Work

Traditional methods for entity and relation extraction usually adopt a pipeline learning framework in which entity and relation extraction tasks are treated as two independent and sequential tasks. Although this framework is flexible, it has limitations such as the possibility of errors in named entity recognition affecting the accuracy of subsequent relation extraction tasks, and the neglect of the inherent connections between entities and relationships, leading to insufficient semantic expression in single-task modeling, which restricts model performance [9,20]. Recently, joint extraction methods have been widely used, which use a unified model for learning and extracting entities and relationships. This approach can integrate the unique information of entities and relationships, enriching the features used in both sub-tasks, and training entity recognition and relationship extraction tasks simultaneously can facilitate optimization and improve the performance of both tasks [6].

There are three commonly used joint extraction models: sequence labeling models, pointer network models, and span-based models. Sequence labeling models are commonly used in named entity recognition and convert the joint extraction task into a sequence labeling prediction task, allowing for the extraction of entities and relations in a single model. Yuan et al. [10] used a relation-aware attention mechanism to construct specific sentence representations for each relation, and then performed sequence labeling to extract their corresponding head and tail entities. Zheng et al. [5] proposed a new labeling scheme and an end-to-end model with a biased objective function to jointly extract entities and their relations. However, this model cannot handle the problem of entity overlap.

The pointer network is a more complex sequence labeling model that uses multiple annotation sequences to represent entities in a sentence and classifies the relationships between entities, thereby solving the problem of entity–relation overlap. Park et al. [21] proposed a relation extraction model based on a double pointer network with multi-head attention mechanism, using the forward decoder of an object decoder to find n-to-1

subject–object relationships and the backward decoder of a subject decoder to find 1-to-n subject–object relationships. Mukherjee et al. [11] adopted an encoder–decoder architecture with a pointer network-based decoding framework, which generates a complete opinion triplet at each time step. The decoder captures the interaction between aspects and opinions by considering their entire detection span while predicting their connection sentiment. On the other hand, Guo et al. [12] proposed a novel BERT-based enhanced lexical adapter (BLA) model, which deeply integrates external dictionary features into the pre-trained language model BERT. However, the pointer network is vulnerable to imbalanced labels, which affects its performance, and its decoding process is complicated, requiring a special program to determine entities based on the pointer network results.

Span-based models identify all possible word spans in text and construct a classification module to predict whether a span is an entity and whether there is a relationship between entities. This type of model can directly obtain entities and relations, solving the problem of entity–relation overlap, and does not require a complex transformation process from labels to entities, making it currently the more advantageous method for joint entity–relation extraction. Several span-based models have been proposed, such as DyGIE [6], which uses a dynamically constructed span graph to share span representations, propagate coreference relationships and relation types in the dynamic graph, and facilitate iterative updates of the span representations. DyGIE++ [7] extended the DyGIE model by integrating global and local information to assist tasks, and uses BERT to construct span representations, while SpanProto [16] introduced global boundary matrices to learn span boundary information and uses prototype learning and margin-based learning to train the model. Ji et al. [17] proposed a span-based joint extraction framework that generates semantic representations for specific spans and contexts based on attention. Ye et al. [18] proposed a new span representation method called the Padding and Layered Marker (PL-Marker), which strategically wraps markers in the encoder to consider the relationships between spans, especially using a neighborhood-based packing strategy to model entity boundary information more effectively. Shen et al. [22] constructed a memory module to remember the category representations learned in entity recognition and relation extraction tasks, and design a multi-level memory flow attention mechanism to enhance bidirectional interaction between entity recognition and relation extraction. These models perform entity and relation classification as two-stage subtasks in a single model. However, enumerating all spans could cause a problem of sample imbalance, as most sentences likely contain few relations between entities. SpERT, a span-based model proposed by Eberts et al. [14], addresses this issue by sampling spans before entity classification and generating negative samples that are strongly negatively related to the positive samples, thereby balancing the training data.

However, the approach described above overlooks the impact of lexical information on the span-based models. Wei et al. [23] proposed a method that considers the advantages of Bi-LSTM in capturing bidirectional semantic dependencies and the attention mechanism in assigning different weights to different parts-of-speech features of words, and combines them to perform entity–relation extraction. Jiang et al. [24] proposed a BERT-BiLSTM-CRF model that integrates part-of-speech (POS) features and regularization methods (BBCPR). However, lexical features and the structures with multi-head self-attention, such as transformer encoder, have not been utilized by span-based models. This may enhance the rich semantic expression of spans based on the advantage of resolving overlapping entities in a span-based model and improve the performance of entity and relationship recognition by filtering effective information through multi-head self-attention. In this paper, we propose a method that enhances the span using multi-level lexical features. Additionally, we introduce an attentional sequential learning structure based on the transformer encoder in the relationship extraction process, which enables the adaptive capturing of intrinsic connections and contextual information between span pairs, improving performance of ERE tasks.

### 3. Methods

The structure of the proposed span-based joint entity and relation extraction model with multi-level lexical and attentional context features (ER-LAC) is shown in Figure 1. The proposed model consists of three parts: a multi-level lexical feature extraction module, an entity classification module, and a relation classification module with attentional context features. The multi-level lexical feature extraction module extracts the part-of-speech (POS) tags of words based on the original text, and then encodes the POS tags that frequently occur in entities and relationships. The POS tags are then fused at both the word and span levels into the original span embedding representation. The enhanced span representation is then classified in the entity classification module to recognize named entities. The recognized named entities are then input into the relation classification module, which uses a transformer structure to extract contextual relationships with attention within and between entities, and extracts relations between entities.

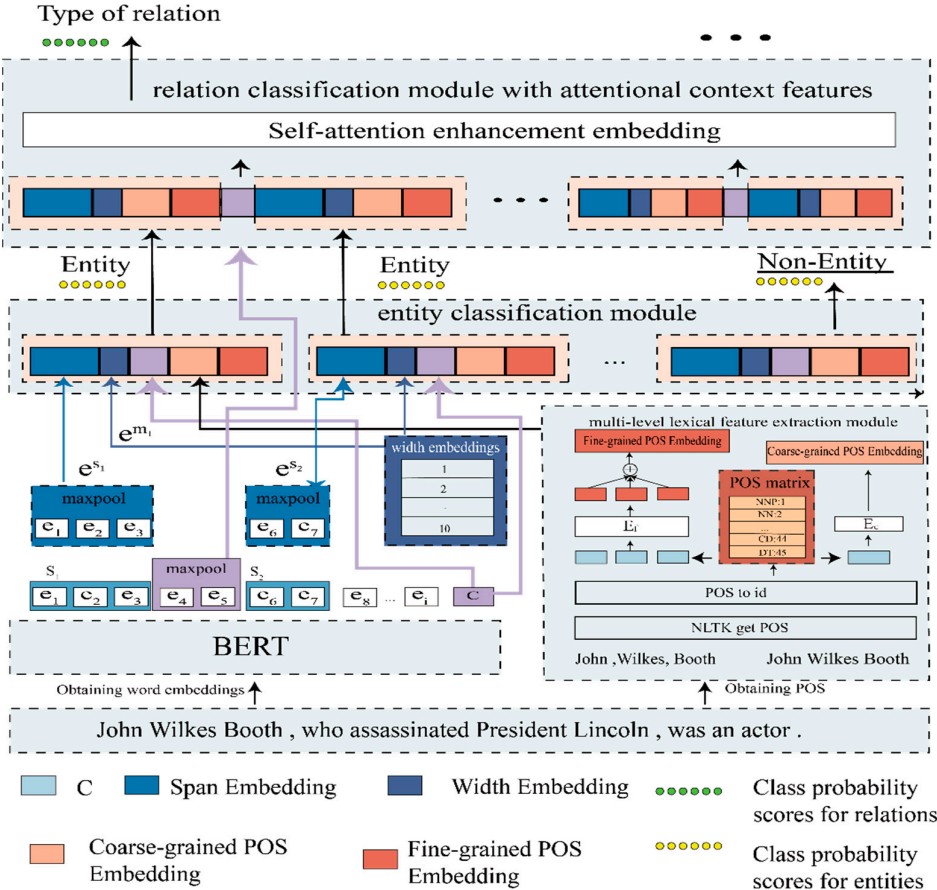

**Figure 1.** The BERT pre-trained model obtained token embeddings and a sentence embedding C. Two types of POS embeddings were proposed to enhance the span embeddings. The enhanced span embeddings were used in the entity classification stage. The embeddings of span pairs and a center embedding with local information were used in the multi-label relation classification stage. The relation classifier used a transformer encoder to extract internal connections between spans.

#### 3.1. Multi-Level Lexical Features Extraction Module

The multi-level lexical feature extraction module first acquires the key POS tags from the original text and encodes each POS tag into an embedding, which is only related to the POS tags. At the same time, a pre-trained language model, BERT, is used to obtain word embeddings for each word. The POS tag embedding representation is then converted into fine-grained and coarse-grained feature representation vectors at the word and span

levels, respectively, and combined with the word embeddings to form a span embedding representation with multi-level lexical feature enhancement.

### 3.1.1. Acquisition of POS Tags

The CoNLL04 [25] dataset was used as a basis for extracting and analyzing the part-of-speech (POS) tags highly associated with named entities, and a POS tag table was determined for POS tag embedding based on the analysis results. The number of corresponding words, entities, and related entities for different POS tags in the CoNLL04 dataset were extracted using the NLTK toolkit [26], as shown in Table 1.

**Table 1.** The numbers of word tokens, entities and entities with relations related to different POS tags in the CoNLL04 dataset.

| Type of the POS Tag | Meaning of the POS Tag | Word Tokens | Entities | Entities with Relations |
|---|---|---|---|---|
| NNP and NNPS | Proper noun | 3045 | 1125 | 868 |
| JJ | Adjective | 308 | 96 | 73 |
| IN | conjunctions | 113 | 17 | 10 |
| VB | Verb | 123 | 40 | 26 |
| RB | Adverb | 37 | 26 | 18 |
| DT | Determiner | 73 | 16 | 9 |
| NN and NNS | Noun | 5061 | 3575 | 2914 |
| CD | Cardinal number | 503 | 253 | 10 |
| Others | Other parts of speech | 279 | 15 | 10 |

It can be observed from the statistical results in Table 1 and the case depicted in Figure 2, that nouns or nouns with modifiers are mostly present in named entities and those with relationships. The POS tags, such as proper nouns singular (NNP), proper nouns plural (NNPS), nouns singular or plural (NN) or nouns plural (NNS), etc., are often observed. Therefore, it can be concluded that the lexical feature could be crucial prior knowledge for entity and relationship extraction tasks. Forty-five types of POS tags were used in the enhanced span embedding in this work, such as NNP, NNPS, NN, NNS and others.

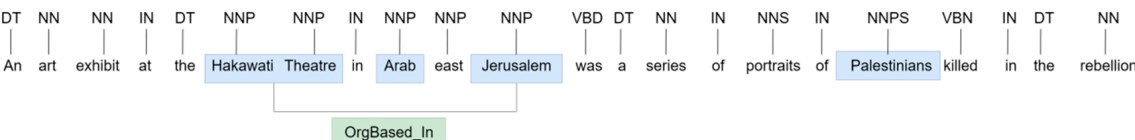

**Figure 2.** An example of lexical properties. The line above the sentence shows the POS annotations of each word in the sentence. In the sentence, the entities were highlighted in blue. The relation was highlighted in green. As can be seen, the POS tags of words in most entities were NNP or NN.

To incorporate POS tags into span representation embedding, the NLTK tool is used to extract the POS tags of word tokens and spans, and then the POS tags are converted into embedding representation. The POS tag of a token or a span is defined by:

$$P_{t_i} = \text{POS}(t_i), P_{s_i} = \text{POS}(s_i) \tag{1}$$

POS is the POS tag recognition function provided by the NLTK toolbox. $P_{t_i}$ is the POS tag of the ith word token $t_i$. $P_{s_i}$ is the POS tag of the ith span $s_i$, which is the combination of word tokens in the span:

$$s_i = \{t_j, t_{j+1}, \ldots, t_{j+m-1}\} \tag{2}$$

These POS tags are then converted into two types of lexical feature embeddings at span-level and token-level, then transformed into coarse-grained and fine-grained POS embeddings, respectively, where they are used to enhance the span embedding at two levels.

### 3.1.2. Coarse-Grained and Fine-Grained POS Embedding

Coarse-grained POS embedding is an $N_c$ dimensional embedding representation obtained by converting the POS tag at the span level. It is calculated by embedding the POS tag by Equation (3):

$$e_i^c = E_c(P_{s_i}), e_i^c \in \mathbb{R}^{N_c} \tag{3}$$

$e_i^c$ is the $N_c$ dimensional coarse-grained POS embedding of span $s_i$. $E_c$ is the trainable embedding function that is commonly used in deep learning. $E_c$ maps a POS tag ID into a $N_c$ dimensional vector. $N_c$ is the dimension of the coarse-grained POS embedding, which is a hyperparameter.

The lexical information of each word within a span is calculated as the fine-grained POS embedding at token level. It calculates the embedding vector of each word POS tag in a span by Equation (4):

$$e_j^t = E_f\left(P_{t_j}\right), e_i^t \in \mathbb{R}^{N_f} \tag{4}$$

$e_j^t$ is the POS embedding of token $t_j$, which converted the POS tag $t_j$ into a $N_f$ dimensional vector. $N_f$ is a hyperparameter. However, each span contains a different number of POS embeddings of tokens, which cannot be simply concatenated to form the embedding vector of a span. Therefore, a weighted sum embedding vector was calculated by these embeddings of tokens to form the fine-grained POS embedding vector of a span, by Equation (5):

$$e_i^f = \sum_{k=j}^{j+m-1} w_k e_k^t, e_i^f \in \mathbb{R}^{N_f} \tag{5}$$

$e_i^f$ is the $N_f$ dimensional fine-grained POS embedding vector of span $s_i$. $w_k$ is the trainable weight for token k in the span.

### 3.2. Entity Classification Module

### 3.2.1. Acquisition of Raw Span Embedding

The text features of input sentences are calculated with n word tokens $D = \{t_1, t_2, \ldots t_n\}$ by Equation (6):

$$\{e_1, e_2, \ldots e_n, C\} = \text{BERT}(D), e_i \in \mathbb{R}^{N_d} \tag{6}$$

BERT is the pre-trained BERT model. $e_i$ is the raw word embedding calculated by BERT. C is the sentence embedding. $N_d$ is the dimension of the word vector output by BERT. The maximum pooling function is used to obtain the raw span embedding vector $e_i^s$, calculated by Equation (7):

$$e_i^s = \text{maxpool}\left(e_j, e_{j+1}, \ldots e_{j+m-1}\right) \tag{7}$$

$e_j, \ldots, e_{j+m-1}$ are the word embeddings of word tokens $t_j, t_{j+1}, \ldots, t_{j+m-1}$ in span $s_i$.

### 3.2.2. Lexical Feature Enhanced Entity Classification

The span embedding representation utilized in the entity extraction stage of the proposed model is composed of the raw span embedding, a sentence embedding, a width embedding, coarse-grained POS embeddings, and fine-grained POS embeddings. Coarse-grained and fine-grained POS embeddings are used to enhance and highlight the part-of-speech features in the span. The enhanced span feature is shown in Figure 3. For example, for a span "Hakawati Theatre", the POS of span "NNP" is obtained using NLTP toolbox, an embedding layer $E_c$ is used to convert the POS of span into coarse-grained POS embedding $e^c$. Then, the POS tags of words in spans are obtained, "NNP" for "Hakawati" and "NNP" for "Theatre". The POS embedding of each POS of word token $e^t$ is converted by embedding layer $E_f$. Next, the fine-grained POS embedding is obtained from $e^t$ using Equation (5). The enhanced embedding of span is composed of $e^f$, $e^c$, the raw span embedding, a sentence embedding and a width embedding.

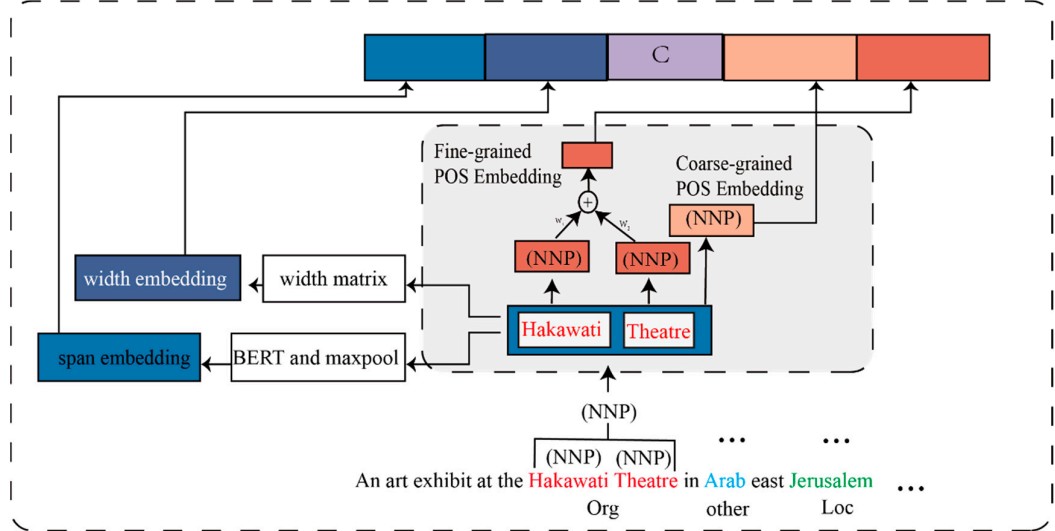

**Figure 3.** The span for entity classification mainly consists of five parts: first, the basic span embedding obtained through maximum pooling (blue), then the width feature of the span (deep purple), sentence vector C as the classification vector for entity classification (light purple), fine-grained part of speech embedding obtained by calculating the part of speech feature of the word vector within the span (orange), and coarse-grained part of speech embedding (brown) obtained by calculating the part of speech features of the span.

In entity classification, the width of the span was embedded by Equation (8):

$$e_i^m = E_m(m_i), e_i^m \in \mathbb{R}^{N_m} \tag{8}$$

$e_i^m$ is the width embedding. $E_m$ is the embedding function for width. $m_i$ is the width of span $s_i$, which is the number of words in the span $s_i$. $N_m$ is the dimension of width embedding which is a hyperparameter. All the embeddings of a span were concatenated and form the span vector $a_i^s$ for classification by Equation (9):

$$a_i^s = \mathrm{concat}\left(e_i^s, e_i^m, C, e_i^c, e_i^f\right), a_i^s \in \mathbb{R}^{N_s} \tag{9}$$

$a_i^s$ is the span embedding enhanced by the lexical feature used for named entity recognition, which contains the raw span embedding $e_i^s$, the width embedding $e_i^m$, the sentence embedding C, and two POS embeddings ($e_i^c$ and $e_i^f$). $N_s = N_d * 2 + N_m + N_c + N_f$.

Finally, a fully connected layer with softmax function is used to calculate the probability of entity for a span using Equation (10):

$$\hat{y}_i^e = \mathrm{softmax}(Wa_i^s + b) \tag{10}$$

$\hat{y}_i^e$ is the probability of predicting to be an entity for span $s_i$. W and b are the trainable weight and bias in the fully connected layer.

After the entity classification, spans that were predicted not to be entities were filtered and did not enter the following relation classification.

### 3.3. Relation Classification Module

After the named entity recognition, span pairs were selected from the set, S, of spans predicted to be entities for relation classification. Since relations have asymmetry, the input span pair $s_i$, $s_j$ is different from span pair $s_j$, $s_i$ for the relation classification.

In the relation classification, a context embedding vector is used for extracting the local context information between span $s_i$ and span $s_j$, which is calculated by Equation (11):

$$c_{i,j} = \mathrm{maxpool}(e_k, e_{k+1}, \dots, e_l), l \geq k \tag{11}$$

$c_{i,j}$ is the context embedding vector between span $s_i$ and $s_j$. When span $s_i$ and $s_j$ overlap, we define $c_{i,j} = 0$. $e_k, e_{k+1}, \ldots, e_l$ are the embeddings of word tokens between the span pair.

The span embedding used in the relation extraction is calculated by Equation (12), where $N_1 = N_f * 2 + N_d + N_m$:

$$a_i^r = \text{concat}\left(e_i^s, e_i^m, e_i^c, e_i^f\right), e_i^r \in \mathbb{R}^{N_1} \tag{12}$$

The span embeddings for relation extraction of the two spans $a_i^r, a_j^r$ and the context embedding vector are concatenated and used in relation classification. The bidirectional relation embeddings $a_{i,j}^r$ and $a_{j,i}^r$ are composed of two span embeddings and the context embedding vector, where $N_a = N_1 * 2 + N_d$:

$$a_{i,j}^r = \text{concat}\left(a_i^r, c_{i,j}, a_j^r\right), a_{i,j}^r \in \mathbb{R}^{N_a}$$
$$a_{j,i}^r = \text{concat}\left(a_j^r, c_{i,j}, a_i^r\right), a_{j,i}^r \in \mathbb{R}^{N_a} \tag{13}$$

Then the transformer encoder is used for classifying the relation embeddings, shown in Figure 4. Relation classifiers use a transformer encoder with the sigmoid function to calculate the probability of each relation type.

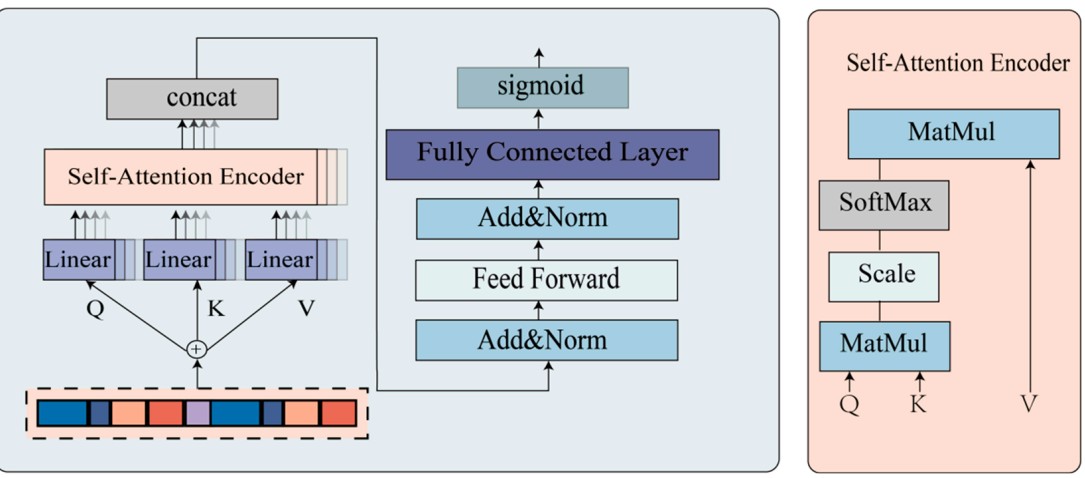

**Figure 4.** Structure of the Multi-head Self-Attention encoder and the self-attention encoder. The colors in the left part of the image correspond to the elements in Figure 1.

The intrinsic connection between span-pair embeddings is captured and learned using a multi-head self-attention layer, which contains multi-head self-attention encoder, and in the experiments, the number of self-attention encoder is set to H = 4. Each self-attention encoder contains three learnable parameters WQ, WK, WV, used to obtain Q, K and V. Each of the three fully connected layers are defined as learnable parameters.

$$Q_{i,j,h}^r = WQ_h\left(a_{i,j}^r\right), K_{i,j,h}^r = WK_h\left(a_{i,j}^r\right), V_{i,j,h}^r = WV_h\left(a_{i,j}^r\right) \tag{14}$$

h denotes the h-th self-attentive encoder. The result of multiplying the query matrix Q with K is divided by the dimension of the input $a_{i,j}^r$ to obtain the scaled dot product attention. Afterwards, softmax transforms probability scores, which are multiplied with the matrix V to obtain the output weight $Z_{i,j,h}^r$ for each self-attention layer.

$$Z_{i,j,h}^r = \text{softmax}\left(\frac{Q_{i,j,h}^r \times K_{i,j,h}^{r \top}}{\sqrt{N_a}}\right)V_{i,j,h}^r \tag{15}$$

where $N_a$ is the feature dimension of input $a_{i,j}^r$. The output $Z_{i,j,h}^r$ obtained from each self-attention layer is stitched together to obtain the output $Z_{i,j}^r$ of the multi-head self-attention layer:

$$Z_{i,j}^r = a_{i,j}^r \cdot \text{concat}\left(Z_{i,j,1}^r, Z_{i,j,2}^r, \ldots Z_{i,j,H}^r\right), Z_{i,j}^r \in \mathbb{R}^{N_a} \tag{16}$$

The output is processed using residual joins as well as layer normalization.

$$O_{i,j}^r = \text{Layer norm}\left(a_{i,j}^r + Z_{i,j}^r\right), O_{i,j}^r \in \mathbb{R}^{N_a} \tag{17}$$

Considering that the attention mechanism is not sufficient for fitting complex processes, a feedforward connection layer is used to enhance the model fitting ability. After the feedforward layer, the final output was obtained by using residual connections again, and layer normalization.

$$E_{i,j}^r = \text{Layer norm}\left(\text{feed forward}\left(O_{i,j}^r\right) + O_{i,j}^r\right), E_{i,j}^r \in \mathbb{R}^{N_a} \tag{18}$$

The predicted results of the relation classifier are defined in Equation (19):

$$\hat{y}_{i,jk}^r = \sigma\left(T\left(WE_{i,j}^r + b\right)\right), 1 \leq k \leq T_n \tag{19}$$

where T denotes the the transformer encoder. $\hat{y}_{i,jk}^r$ is the predicted probability of the kth relation from span $s_i$ to $s_j$. $\sigma$ is the sigmoid function. $N_r$ is the number of relation types. If all the $\hat{y}_{i,jk}^r$ for $N_r$ relations were below threshold (generally 0.5), the model would predict no relation from $s_i$ to $s_j$. If any $\hat{y}_{i,jk}^r$ was above the threshold, the model would select all kth relations as the predicted relations.

### 3.4. Loss Function

We define the loss function L as the sum of the entity classification loss function and the relation classification loss function. The loss function $L_{ner}$ for entity classification is the cross-entropy and the loss function $L_{rel}$ for relation classification is the average value of binary cross-entropy for each relation type, calculated by Equations (20)–(22).

$$L_{ner} = -\frac{1}{N_{ner}}\sum_{i=1}^{N_e} y_i^e \ln \hat{y}_i^e \tag{20}$$

$$L_{rel} = -\frac{1}{N_{rel}}\sum_{i=1}^{N_r} y_i^r \log \hat{y}_i^r + (1 - y_i^r)\log\left(1 - \hat{y}_i^r\right) \tag{21}$$

$$L = L_{ner} + L_{rel} \tag{22}$$

$N_{ner}$ and $N_{rel}$ represent the number of samples of entities and relations.

## 4. Discussion

### 4.1. Datasets and Experiment Setups

The proposed model was evaluated in three commonly used public ERE datasets: CoNLL04 [25], ADE [27], and SciERC [1].

The CoNLL04 dataset contains sentences with annotated named entities and relations extracted from news articles. It contains a training set consisting of 1153 texts and a test set consisting of 288 texts written in English. The dataset comprises four entity types (persons, locations, organizations, and others) and five relation types (Work-For, Kill, OrganizationBased-In, Live-In, and Located-In). The partition of the CoNLL04 dataset that was used for testing SpERT [14] and DyGIE++ [7] is utilized in the experiments.

The ADE dataset contains Medline case reports in which drugs, adverse effects, doses, and their relations were annotated. Entities and relations are systematically annotated to ensure data quality. Experiments were conducted using ten-fold cross-validation with a

total sample size of 4272. For each validation, 3845 samples were used for training and the rest of the 427 samples were used for testing.

The SciERC dataset includes annotations of scientific entities, the relations between them, and co-citation clusters for 500 scientific abstracts. These abstracts are extracted from the AI conference/workshop proceedings. We used the same dataset partition as in [1], containing a training set (1861 sentences), a development set (275 sentences) and a test (551 sentences). For the actual training, we used both the training set and the development set together for training. The details of the three datasets are shown in Table 2.

**Table 2.** The details of the three datasets.

| Datasets | Entities | Relations | Total Number of Samples | Cross-Validation |
|----------|----------|-----------|-------------------------|------------------|
| CoNLL04  | 5349     | 2048      | 1441                    | NO               |
| SciERC   | 8089     | 4716      | 2687                    | NO               |
| ADE      | 10,839   | 6821      | 4272                    | YES              |

In the experiments, we evaluated the proposed model using the predefined training and test datasets in CoNLL04 and SciERC. Additionally, the model was evaluated in the ADE dataset using ten-fold cross-validation. To test the performance of the proposed lexical features and the transformer encoder for relation classification, ablation experiments were conducted based on a basic span-based ERE model SpERT in the CoNLL04 dataset [14]. The proposed model was compared with several state-of-the-art (SOTA) entity–relation extraction models on the three datasets. The performance of each model was evaluated using precision, recall, and F1 scores in both ablation experiments and comparisons with other models, and the advantages of the proposed method were analyzed as well. However, the proposed method has more added features and network structures, which may result in additional computational complexity. To ensure that the proposed model did not have too much of an increase in computational complexity compared to the basic SpERT model, a computational complexity analysis of the proposed method was conducted.

The hyperparameters were manually optimized and set as follows: batch size = 2, learning rate = $5e - 5$, number of epochs = 20, $N_r = 5$, m is the width of spans, $1 \leq m \leq 10$, $N_c = 100$, $N_f = 50$, $N_m = 20$. The hyperparameters used in the proposed model were manually optimized and set to the same values as in SpERT, and the AdamW optimizer was used. The experiments were implemented on a workstation with 24 CPU cores, 64G RAM, one NVIDIA RTX 3090 GPU, CentOS 7.5, Python 3.6, and Pytorch 1.9.0.

*4.2. Experiment Result*

4.2.1. Ablation Experiments

Ablation experiments were conducted on the CoNLL04 dataset to verify the effectiveness of multi-level lexical features and the transformer encoder in relation classification. The SpERT model was used as the baseline model, and the proposed features and classifiers were added to the SpERT model to evaluate the effectiveness of the proposed methods. All gains in the experiments were absolute gains. Six different models were built as follows, based on the SpERT model and adding the proposed features or classifiers:

1. SpERT+E$\left(e_i^c\right)$: SpERT model with coarse-grained POS embedding in the entity classification;
2. SpERT+E$\left(e_i^f\right)$: SpERT model with fine-grained POS embedding in the entity classification;
3. SpERT+E$\left(e_i^c + e_i^f\right)$: SpERT model with coarse-grained and fine-grained POS embedding in the entity classification;
4. SpERT+R$\left(e_i^c\right) + E\left(e_i^c + e_i^f\right)$: Model 3 with coarse-grained POS embedding in the relation classification;
5. SpERT+T: SpERT model with the transformer encoder;
6. SpERT+R$\left(e_i^c\right) + E\left(e_i^c + e_i^f\right) + T$ Model 4 with the transformer encoder.

The results of the ablation experiments are shown in Table 3, and all evaluation metrics in the experiment were calculated using macro-averaged values. It can be seen from the

analysis of models 1 and 2 that the performance of the entity recognition and relation extraction subtasks can be improved by adding fine-grained and coarse-grained POS features. The impact of coarse-grained POS features is greater, and the F1 scores of entity and relation extraction are increased by 0.85% and 0.99%, respectively, compared with the baseline model. The fine-grained POS features improved the F1 scores of entity recognition by 0.31% and 0.36% relative to the baseline model. In model 3, both fine-grained and coarse-grained POS features were added in the entity–relation extraction subtask, and the results showed that adding both features to the entity subtask can improve the performance of entity recognition and relation extraction. The F1 scores were increased by 0.95% and 1.31%, respectively, compared with the baseline model. In model 4, coarse-grained POS features were added in the relation classification subtask based on model 3. The results showed that the F1 scores of model 4 were increased by 0.25% and 0.69% relative to model 3, and by 1.2% and 2.00% relative to the baseline model. This indicates that the proposed POS features can not only improve the performance of the overall task in the entity recognition subtask, but also have a promotion effect on the overall performance of the model in the relation extraction subtask. As the impact of fine-grained POS features is small, fine-grained POS features were not added separately in relation classification.

**Table 3.** Results of ablation experiments on the dataset CoNLL04. The SpERT model was used as the baseline model. The macro precision, recall and F1-scores were calculated for named entity recognition (NER) and relation extraction (RE) subtasks. The bolded portion indicates the best result for each column.

| Experiment | NER (%) | | | RE (%) | | |
|---|---|---|---|---|---|---|
| | Precision | Recall | F1-Score | Precision | Recall | F1-Score |
| SpERT | 85.78 | 86.84 | 86.25 | 74.75 | 71.52 | 72.87 |
| SpERT+E$(e_i^c)$ | 88.47 | 85.98 | 87.06 ($\pm$0.12) | 73.38 | **75.22** | 73.86 ($\pm$0.14) |
| SpERT+E$(e_i^f)$ | 87.06 | 86.23 | 86.56 ($\pm$0.28) | 76.33 | 70.89 | 73.23 ($\pm$0.50) |
| SpERT+E$(e_i^c + e_i^f)$ | 87.34 | 87.17 | 87.20 ($\pm$0.14) | 75.70 | 72.98 | 74.18 ($\pm$0.11) |
| SpERT+R$(e_i^c)$ + E$(e_i^c + e_i^f)$ | **89.33** | 86.00 | 87.45 ($\pm$0.27) | 75.46 | 74.56 | 74.87 ($\pm$0.10) |
| SpERT+T | 88.60 | 86.23 | 87.28 ($\pm$0.56) | 74.88 | 73.46 | 74.16 ($\pm$1.03) |
| SpERT+R$(e_i^c)$ + E$(e_i^c + e_i^f)$ + T | 87.81 | **88.80** | **88.29 ($\pm$0.22)** | **75.72** | 74.91 | **75.13 ($\pm$0.33)** |

Model 5 added a transformer encoder for classification in the relation classification task compared to the baseline model. The results showed that adding a transformer encoder alone can increase the F1 scores of entity recognition and relation extraction by 1.04% and 1.29%, respectively. Model 6 added a transformer encoder in the relation classification task based on model 5, which fully utilized the multi-level lexical features proposed in this paper and the transformer encoder in relation classification. The results showed that the F1 scores of model 6 were increased by 1.01% and 0.97% compared with model 5, and by 2.04% and 2.26% relative to the baseline model. This indicates that the transformer encoder model in the relation classification task can not only improve the baseline model alone, but also have a significant improvement effect on the performance of the baseline model when combined with POS features. The results of the above ablation experiments also show that the lexical features and the transformer encoder in the relation classification proposed in this paper are important for entity–relation extraction tasks, which can greatly improve the comprehensive performance of the span-based entity–relation joint extraction model.

Figure 5 shows the confusion matrix of the final model tested on the dataset CoNLL04, with the baseline results, including entity classification as well as relation classification. It can be seen that our proposed ER-LAC model, model 6 in the ablation experiments, has better classification results than the base model for both subtasks, which illustrates the clear advantage of our proposed method in the entity–relation extraction task.

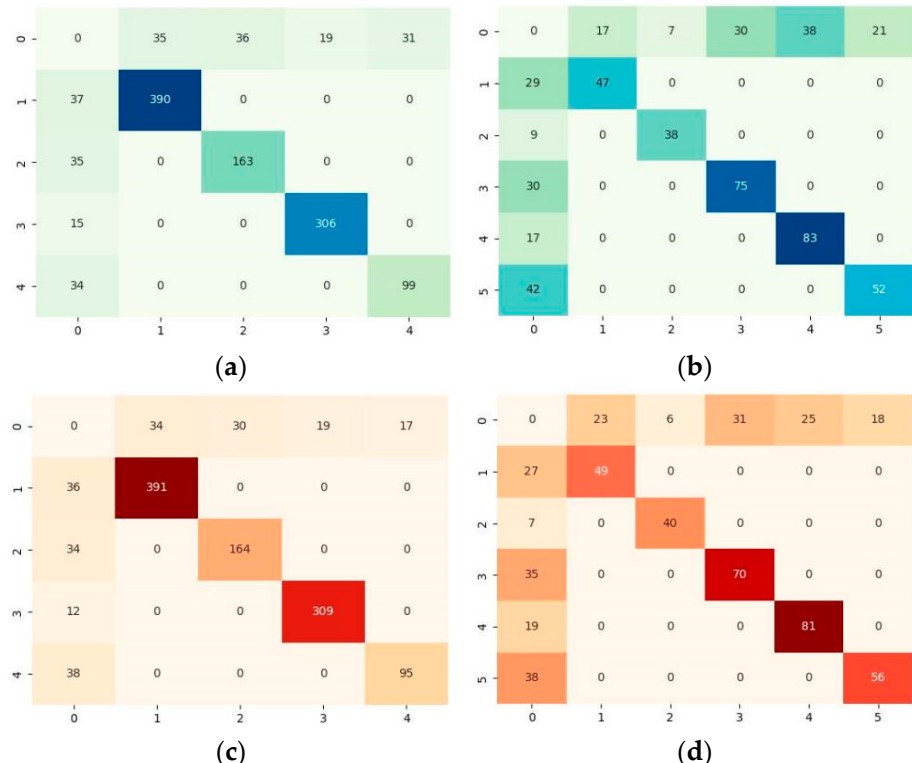

**Figure 5.** The confusion matrix of the models tested on dataset CoNLL04, where (**a**) is the entity classification result of the baseline model, (**b**) is the relation classification result of the baseline model, (**c**) is the entity classification result of our proposed model, and (**d**) is the relation classification result of our proposed model.

### 4.2.2. Comparison with Other Models

The proposed ER-LAC model was also compared with other SOTA models on three datasets. The results with the CoNLL04 dataset are shown in Table 4. Since other models reported the performance of the model using micro-averaging and macro-averaging metrics, we calculated the micro-averaged and macro-averaged evaluation metrics of the proposed model in the entity recognition and relation extraction subtasks. It can be seen that the proposed model can achieve the highest F1 score in the relation extraction subtask, with a macro F1 score of 75.13% and a micro F1 score of 73.04%, which are 2.26% and 1.57% better than the best-performing SOTA models, respectively. The micro F1 score of the proposed model in the entity recognition subtask reached the highest at 90.47%, which is 0.69% better than the best-performing model. Although the macro F1 score of entity recognition did not reach the highest level, it was second only to the TriMF model [22]. However, the same model has a much higher score in relation extraction than other models, indicating that the reason for the proposed model's macro F1 score not reaching the highest in entity recognition may be due to the fact that the entity–relation joint extraction task is more biased towards relation extraction.

Table 5 shows the results of the proposed model and other SOTA models on the ADE dataset. The evaluation metrics for all models were based on macro-averaged values. It can be seen that the proposed model achieved the highest F1 scores in both entity recognition and relation extraction subtasks, with scores of 92.34% and 84.86%, respectively. The F1 score of the proposed model improved by 0.84% to 5.23% relative to the SOTA models in the entity recognition subtask, and by 0.96% to 7.57% in the relation extraction subtask. Moreover, the proposed model achieved the highest precision and recall in the entity recognition, as well as precision in the relation extraction.

**Table 4.** Performance of the proposed model and other SOTA models on the CoNLL04 dataset. The bolded portion indicates the best result for each column.

| Model | CoNLL04 | | | | | |
|---|---|---|---|---|---|---|
| | NER (%) | | | RE (%) | | |
| | Precision | Recall | F1 | Precision | Recall | F1 |
| Global [28] | - | - | 85.60 | - | - | 67.80 |
| Multi-turn QA [29] | 89.00 | 86.60 | 87.80 | 69.20 | 68.20 | 68.90 |
| SpERT [14] | 85.78 | 86.84 | 86.25 | 74.75 | 71.52 | 72.87 |
| SpERT [14] * | 88.25 | 89.64 | 88.94 | 73.04 | 70.00 | 71.47 |
| Deeper [30] | 89.72 | 86.42 | 87.00 | 77.73 | 68.38 | 72.63 |
| Deeper [30] * | 89.84 | 89.73 | 89.78 | **78.69** | 64.84 | 71.08 |
| TriMF [22] | **90.26** | 90.34 | 90.30 | 73.01 | 71.63 | 72.35 |
| ER-LAC | 87.81 | 88.80 | 88.29 | 75.72 | **74.91** | **75.13** |
| ER-LAC * | 89.93 | **91.01** | **90.47** | 74.57 | 71.56 | 73.04 |

* The reported performance is a micro-average.

**Table 5.** Performance of the proposed model and other SOTA models on the ADE dataset. All the metrics use macro-averaged values. The bolded portion indicates the best result for each column.

| Model | ADE | | | | | |
|---|---|---|---|---|---|---|
| | NER (%) | | | RE (%) | | |
| | Precision | Recall | F1 | Precision | Recall | F1 |
| Relation-Metric [31] | 86.16 | 88.08 | 87.11 | 77.36 | 77.25 | 77.29 |
| CLDR+CLNER [32] | - | - | 88.3 | - | - | 79.97 |
| SpERT [14] | 88.99 | 89.59 | 89.28 | 77.77 | 79.96 | 78.84 |
| Table-Sequence [33] | - | - | 89.70 | - | - | 80.10 |
| CMAN [34] | - | - | 89.40 | - | - | 81.14 |
| REBEL [35] | - | - | - | - | - | 82.20 |
| Deeper [30] | 89.06 | 89.63 | 89.48 | 80.51 | 86.81 | 83.74 |
| PFN(ALBERT) [36] | - | - | 91.50 | - | - | 83.90 |
| ER-LAC | **91.67** | **93.03** | **92.34** | **83.07** | **86.74** | **84.86** |

Table 6 shows the results of the proposed model and other SOTA models on the SciERC dataset. All models' evaluation metrics use micro-averaged values. The proposed model achieves the highest F1 score in entity recognition, reaching 70.72%, which is an improvement of 0.19–5.52%, compared to other models. In the relation extraction subtask, the ER-LAC model is second only to the PL-Marker model, achieving the second highest F1 score. However, the proposed model performs better than the PL-Marker model in entity recognition. Therefore, this result may also be caused by the bias in the training of the two subtasks. Based on this result, it can also be seen that the proposed model has reached the level of the best SOTA model on the SciERC dataset. Overall, the proposed ER-LAC model outperforms, or is on par, with the current best SOTA models in all three public datasets.

### 4.2.3. Complexity Analysis

Although the superior performance of the proposed ER-LAC model has been validated in both ablation experiments and comparison experiments with other SOTA models, the model introduces more feature extraction and selection modules compared to the baseline model SpERT, which may significantly increase the training and inference time and make it difficult to be deployed in practice. Therefore, to demonstrate that the ER-LAC model still has lower resource consumption and higher computational speed, we conducted a complexity analysis on the model, primarily comparing the parameter size, average sample training time, and inference time between ER-LAC and the baseline model. Table 7 shows the results of the complexity analysis experiments. The average sample training time is

the time taken to compute each sample on average during the training process, and the inference time is the time taken to compute each sample on average during the inference process. The results show that the proposed model has increased the parameter size by 9.27% and added less than 5% of the training time and inference time compared to the baseline model, which indicates that the increase in computational cost is minimal and the proposed model still has lower computational consumption, ensuring efficient usage in practical applications.

**Table 6.** Performance of the proposed model and other SOTA models on the SciERC dataset. All the metrics use micro-averaged values. The bolded portion indicates the best result for each column.

| Model | SciERC | | | | | |
|---|---|---|---|---|---|---|
| | NER (%) | | | RE (%) | | |
| | Precision | Recall | F1 | Precision | Recall | F1 |
| PFN [37] | - | - | 69.9 | - | - | 53.2 |
| DyGIE [6] | - | - | 65.2 | - | - | 41.6 |
| DyGIE++ [7] | - | - | 67.50 | - | - | 48.40 |
| Cross-sentence [38] | - | - | 68.90 | - | - | 50.10 |
| SpERT$_{SciBERT}$ $^\Delta$ [8] | **70.87** | 69.79 | 70.33 | 53.40 | 48.54 | 50.84 |
| SpERT.PL$_{SciBERT}$ [39] | 69.82 | 71.25 | 70.53 | 51.94 | 50.62 | 51.25 |
| PL-Marker$_{SciBERT}$ [18] | - | - | 69.90 | - | - | **53.20** |
| ER-LAC$_{SciBERT}$ | 65.26 | **77.18** | **70.72** | 46.51 | **59.70** | 52.29 |

$^\Delta$ indicates that there are overlapping entities in the dataset.

**Table 7.** Complexity analysis.

| Model | Baseline Model (SpERT) | ER-LAC | Increment |
|---|---|---|---|
| The number of parameters | 108,331,782 | 118,383,920 | +9.27% |
| Training time (ms) | 24.10 | 24.36 | +1.07% |
| Inference time (ms) | 10.54 | 11.24 | +6.64% |

## 5. Conclusions and Future Works

In this paper, we propose a span-based joint entity and relation extraction model with enhancement of lexical features and internal connections (ER-LAC). On the basis of using BERT to obtain word embedding and span embedding, we added two types of lexical part-of-speech (POS) features to enrich the embedding representation of span and used transformer to capture the internal connection between span pairs.

We conducted two types of experiments to verify the effectiveness of the proposed method. Firstly, in the ablation experiments using the SpERT model as the baseline model on which the multi-granularity lexical features were verified separately from the ablation experiments, it can be seen that the coarse-grained lexical feature has a relatively obvious improvement for entity extraction. Further enhancing the semantic representation of span, the f1 score of entity classification is significantly improved by 0.95%, while the enhancement of span embedding brings a 1.31% improvement to the f1 score of relation classification, proving that the enhancement of lexical features for span embedding is effective and also very effective for the entity–relation classification task. Secondly, we experimented with the encoder structure of the transformer alone without adding any lexical features to capture the intrinsic connection between span pairs in the relation classification task, and we can see that the f1 score of entity classification is significantly improved by 1.03%, while the enhancement of span embeddings brings a 1.29% improvement in the f1 score of relation classification. Finally, we fused all the innovations into the model to obtain the ER-LAC model with a 2.04% improvement in entity classification f1 score. The f1 score

of relational classification was improved by 2.26%. The effectiveness of the innovation points is proved.

We also conducted experiments on two other datasets, ADE and SciERC, which are more popular in the field of entity relations and compared them with the SOTA model to demonstrate the advancement of our model's innovation points as well as its usability. The experimental results showed that the two types of proposed POS features and internal connection relation classifiers were effective for entity and relation joint extraction. The proposed ER-LAC model also outperformed the other state-of-the-art models in three commonly used datasets. In future studies, it still has great potential for improvement, such as in how to improve the embedded representation of span in ERE tasks.

In our experiments, it can be seen that enhancing the semantic information across the span is a feasible solution for the entity–relation classification task, and in future work, we will also consider adding other features, such as syntactic tree structure information, phrase categories, etc., to enhance the representation capability. Moreover, we will explore more efficient entity–relation classifiers to reduce the additional computation associated with the additional features.

Due to the recent proposal of the Large Language Model, we found a great advantage in the GPT-based model on NLU tasks, and in future work, we will think about how to merge the existing work with the GPT-based model for research innovation.

**Author Contributions:** Conceptualization, Z.W.; Methodology, Y.Z. (Yaqin Zhu); Software, X.L.; Investigation, C.Y.; Data curation, J.L.; Writing—original draft, Y.Z. (Yaqin Zhu); Writing—review & editing, Z.W.; Supervision, Y.Z. (Yanting Zhang), Z.W. All authors have read and agreed to the published version of the manuscript.

**Funding:** This research was funded by National Natural Science Foundation of China (No. 62302090 and No. 62272097), Shanghai Sailing Program (No. 23YF1401100) and Fundamental Research Funds for the Central Universities (No. 2232021D-26).

**Data Availability Statement:** Publicly available datasets were analyzed in this study. This data can be found here: [ADE dataset: https://doi.org/10.1016/j.jbi.2012.04.008, SciERC dataset: Multi-Task Identification of Entities, Relations, and Coreferencefor Scientific Knowledge Graph Construction (washington.edu), CoNLL04 dataset: Performance on test set for NER and RE; RE in pipeline... | Download Scientific Diagram (researchgate.net)].

**Conflicts of Interest:** The authors declare no conflict of interest.

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
