# Peer review of "ER-LAC: Span-Based Joint Entity and Relation Extraction Model with Multi-Level Lexical and Attention on Context Features"

_applsci, doi:10.3390/app131810538_

Round 1

Reviewer 1 Report

This paper develops ER-LAC, a joint entity-relation extraction, and evaluate it on CoNLL04 dataset. Overall the new presented method is explaained clearly with supoorting experimental results.

I have the following questions:

1. Reference format need to be correct.

2. The proposed model outperforms traitional models in accuracy, I am interested in how is its speed, as its structure consists of many modules.

3. I am interested how are GPT-based models perform on this task, as they have been found outperform BERT-based model on NLU tasks too.

Author Response

Thank you to the reviewers for their comments, I have made changes to the manuscript based on the comments, please see the attached content for the response.

Reviewer 2 Report

Overall a solid piece of work. It is generally a conceptual paper with a sufficient coverage of validity of Authors' assumptions via experimental studies.

The paper can be improved editorially a bit by setting to bold face the best results in each column of the result tables.

As the Authors' model has a couople of hyperparameters to be set by the user, it would be insightful to know how the hyperparameters should be set for a concrete dataset (or it should be demonstrated that the results are not so sensitive to the hyperparameters or that the choice done by the Authors is universally applicable).

In any case, the effects of mankpulating the hyperparameters should be exposed. It would of course considerably increase the volume of the paper so I would recommend to present such a study as an accompanying on-line material.

Author Response

Thank you very much for your valuable comments on the thesis, I have made corresponding changes to the thesis according to the comments, please see the attached content for the specific response.
